# Development and validation of multivariable clinical diagnostic models to identify type 1 diabetes requiring rapid insulin therapy in adults aged 18–50 years

Anita Lynam,[1] Timothy McDonald,[1,2] Anita Hill,[1] John Dennis,[1] Richard Oram,[1,3] Ewan Pearson,[4] Michael Weedon,[1] Andrew Hattersley,[1,5] Katharine Owen,[6,7] Beverley Shields,[1] Angus Jones[1,5]

BS and AJ are joint senior authors.

For numbered affiliations see end of article.

**Correspondence to**
Dr Angus Jones;
Angus.Jones@exeter.ac.uk

## ABSTRACT

**Objective** To develop and validate multivariable clinical diagnostic models to assist distinguishing between type 1 and type 2 diabetes in adults aged 18–50.

**Design** Multivariable logistic regression analysis was used to develop classification models integrating five pre-specified predictor variables, including clinical features (age of diagnosis, body mass index) and clinical biomarkers (GADA and Islet Antigen 2 islet autoantibodies, Type 1 Diabetes Genetic Risk Score), to identify type 1 diabetes with rapid insulin requirement using data from existing cohorts.

**Setting** UK cohorts recruited from primary and secondary care.

**Participants** 1352 (model development) and 582 (external validation) participants diagnosed with diabetes between the age of 18 and 50 years of white European origin.

**Main outcome measures** Type 1 diabetes was defined by rapid insulin requirement (within 3 years of diagnosis) and severe endogenous insulin deficiency (C-peptide <200 pmol/L). Type 2 diabetes was defined by either a lack of rapid insulin requirement or, where insulin treated within 3 years, retained endogenous insulin secretion (C-peptide >600 pmol/L at ≥5 years diabetes duration). Model performance was assessed using area under the receiver operating characteristic curve (ROC AUC), and internal and external validation.

**Results** Type 1 diabetes was present in 13% of participants in the development cohort. All five predictor variables were discriminative and independent predictors of type 1 diabetes (p<0.001 for all) with individual ROC AUC ranging from 0.82 to 0.85. Model performance was high: ROC AUC range 0.90 (95% CI 0.88 to 0.93) (clinical features only) to 0.97 (95% CI 0.96 to 0.98) (all predictors) with low prediction error. Results were consistent in external validation (clinical features and GADA ROC AUC 0.93 (0.90 to 0.96)).

**Conclusions** Clinical diagnostic models integrating clinical features with biomarkers have high accuracy for identifying type 1 diabetes with rapid insulin requirement, and could assist clinicians and researchers in accurately identifying patients with type 1 diabetes.

### Strengths and limitations of this study

► Diabetes type is robustly defined using direct measurement of endogenous insulin secretion, an outcome closely related to treatment, education and monitoring requirements.

► A combination of a large development dataset and small number of predictors minimises risk of model overfitting, a common problem with diagnostic models of this nature.

► Models are robustly internally and externally validated.

► The cross-sectional nature of the development and validation cohorts means that time to insulin was self-reported and measurement of model predictors was not undertaken at diagnosis: both body mass index and islet autoantibody prevalence may change over time.

► Models have been developed in white European populations with young adult onset diabetes: further work is required to extend this work to other age groups and ethnicities.

## INTRODUCTION

Making the correct diagnosis of type 1 and type 2 diabetes is crucial for appropriate management, with guidelines for these conditions recommending very different glucose-lowering treatment and education.[1–3] These differences are predominantly driven by the rapid development of severe endogenous insulin deficiency in type 1 diabetes.[1] This means that patients with type 1 diabetes need rapid insulin treatment and are at risk of life-threatening ketoacidosis without insulin treatment. They develop a requirement for physiological insulin replacement (eg, multiple injections, carbohydrate counting, pumps) due to the very high glycaemic variability associated with severe

insulin deficiency[4 5] and have poor glycaemic response to most adjuvant glucose-lowering therapies.[6] In contrast, patients with type 2 diabetes continue to make substantial endogenous insulin even many decades after diagnosis.[7] Glycaemia is therefore usually managed initially with lifestyle change or oral agents[4 8] and, if insulin treatment is needed, a combination of simple insulin regimens and adjuvant non-insulin therapies.[4 5 8 9]

Correctly distinguishing between diabetes subtypes at diagnosis is often difficult and misclassification is therefore common.[10–12] Current guidelines focus on etiopathological definitions without giving clear criteria for clinical use.[1 13] In clinical practice, clinical features are predominantly used to determine diabetes subtype but only age at diagnosis and body mass index (BMI) have evidence for utility at diabetes onset, whereas other features used by clinicians such as symptoms at diagnosis, weight loss or ketosis do not have an evidence base.[14] Increasing obesity rates mean that many patients with type 1 diabetes will be obese and type 2 diabetes is occurring in the young.[15] Type 1 diabetes has been recently shown to occur at similar rates in those aged above and below 30 years.[16] Therefore, simple cut-offs based on age at diagnosis and BMI are unlikely to accurately diagnose diabetes type for many patients.[1 10] Similarly, there is no single diagnostic test that can be used to classify diabetes robustly at diagnosis. While measurement of islet autoantibodies can assist classification, many patients with type 1 diabetes are islet autoantibody negative and many patients with the clinical phenotype of type 2 diabetes, without rapid insulin requirement, are islet autoantibody-positive.[17] A Type 1 Diabetes Genetic Risk score (T1D GRS) has been recently shown to assist diagnosis of diabetes type but this provides imperfect discrimination in isolation.[18]

To classify diabetes, a suitable 'gold standard' is necessary. As the key factor driving differences in treatment decisions between the two subtypes is the lack of endogenous insulin secretion, direct measurement of endogenous insulin secretion in long-standing insulin-treated diabetes (>3–5 years), using C-peptide, provides a robust classification that closely relates to treatment requirements[19]; patients with severe endogenous insulin deficiency (low C-peptide) have the high glucose variability, absolute insulin requirement and lack of response to non-insulin glucose-lowering therapies that are characteristic of type 1 diabetes, regardless of their clinical characteristics and clinician's diagnosis.[7 11 19–23] However, this test may have limited utility at diagnosis, as patients with recent onset type 1 diabetes may have retained endogenous insulin secretion.[21 24]

Clinical prediction models offer a way of combining multiple patient features and biomarkers to improve accuracy of diagnosis or prognosis. In diabetes, diagnostic models combining clinical features are available to predict the risk of prevalent or incident type 2 diabetes[25] and there is a model to identify monogenic forms of diabetes in patients with young-onset diabetes.[26] However, there are no statistical prediction models to help distinguish type 1 and type 2 diabetes at diagnosis. We therefore aimed to develop and validate multivariable clinical diagnostic models that combine clinical features and biomarkers to identify type 1 diabetes (defined by rapid insulin requirement and severe endogenous insulin deficiency) in patients aged between 18 and 50 years at diabetes diagnosis.

## METHODS

We used logistic regression to model the relationship between each of clinical features and biomarkers, and type 1 diabetes defined by rapid insulin requirement and severe endogenous insulin deficiency (see later). We assessed the performance of the models using both internal validation and external validation.

### Study population: development cohort

For model development, participants were identified from Exeter, UK-based cohorts.[27–30] These cohorts were participants with clinically diagnosed diabetes recruited from primary and secondary care. Summaries of the cohorts including recruitment and data collection methods are shown in online supplementary table 1.

Participants were eligible for the study (model development or validation) if they had a clinical diagnosis of diabetes between the ages of 18 and 50 years. Participants with known secondary or monogenic diabetes,[31] or a known disorder of the exocrine pancreas,[32] were excluded. All participants included in this study were of white European origin.

### Study population: external validation cohort

Participants meeting the study inclusion criteria were identified in the Young Diabetes in Oxford (YDX) study.[33] YDX is a cross-sectional study of participants diagnosed with diabetes (of any type) up to the age of 45 years, recruited from primary and secondary care in the Thames Valley region, UK. Participants with known secondary, pancreatic or monogenic diabetes were excluded.

### Model outcome: type 1 and type 2 diabetes definition

Type of diabetes was defined by the presence or absence of rapid insulin requirement and severe endogenous insulin deficiency after a diagnosis of diabetes, as follows:

Type 1 diabetes: Insulin treatment within ≤3 years of diabetes diagnosis and severe insulin deficiency (non-fasting C-peptide <200 pmol/L).[21]

Type 2 diabetes: Either (1) no insulin requirement for 3 years from diabetes diagnosis or (2) where insulin was started within 3 years of diagnosis, substantial retained endogenous insulin secretion (C-peptide >600 pmol/L) at ≥5 years diabetes duration.

Cohort participants not meeting the above criteria or with insufficient information were excluded from analysis, as type of diabetes and rapid insulin requirement could not be robustly defined.

## Model predictors

Five pre-specified predictor variables were assessed, based on prior evidence and availability: age at diagnosis,[14] BMI,[14] GADA (Glutamic acid decarboxylase autoantibody) and IA-2A (Islet antigen-2 autoantibody),[17 34] and a T1D GRS.[18]

## Assessment of clinical features

At study recruitment visit, clinical history including time to insulin and age at diagnosis were self-reported by participants in an interview with a research nurse. Height and weight were measured for calculation of BMI.

## Laboratory measurement
### C-peptide

In the development cohort, C-peptide was measured on stored EDTA taken at study visits (non-fasting random,[35] fasting or at 90 min in a post-mixed-meal tolerance test (majority 87% non-fasting)). With specific additional consent, C-peptide was also measured on post-recruitment non-fasting EDTA (Ethylenediaminetetraacetic acid) samples collected as part of routine clinical care. Fasting C-peptide values were multiplied by 2.5 to non-fasting equivalent.[21] The median C-peptide value was used where more than one eligible C-peptide value was available (62% of participants requiring this measure for outcome definition). C-peptide was measured using an electrochemiluminescence immunoassay on a Roche Diagnostics E170 analyser (Roche, Mannheim, Germany) by the Academic Department of Blood Sciences at the Royal Devon and Exeter Hospital. In the external validation cohort, C-peptide measurement was performed in the Biochemistry Laboratory of the Oxford University Hospitals NHS (National Health Service) Trust using a chemiluminescence immunoassay on an ADVIA Centaur analyser (Siemens Healthcare Diagnostics).

### Islet autoantibodies

In the development cohort, GADA and IA-2A were measured on EDTA taken at recruitment or obtained from local laboratory records. Both islet autoantibodies were measured using the RSR ELISA assays (RSR Ltd, Cardiff, UK) on the Dynex DS2 ELISA Robot (Dynex Technologics, Worthing, UK) by the Academic Department of Blood Sciences at the Royal Devon and Exeter Hospital. The department participates in the International Autoantibody Standardisation Programme. The cut-off for positivity for GADA was ≥11 units/mL and IA-2A was ≥15 units/mL, based on the 97.5th percentile of 1559 controls without diabetes.[34]

In the external validation cohort, GADA was measured by a radioimmunoassay using $^{35}$S-labelled full-length GAD65 by the Department of Clinical Science, University of Bristol, Bristol, UK. Results were expressed in WHO units per millilitre derived from a standard curve calibrated from international reference material (National Institute for Biological Standards and Control code 97/550). The cut-off for positivity for GADA was 13 WHO

Units/mL initially, using a local assay (samples measured n=218, Diabetes Antibody Standardisation Program (DASP) 2010 sensitivity 88% and specficity i93%) and changed to 33 DK units/mL later in the study (standard assay, DASP2010 sensitivity 80%, specificity 97%).

### Type 1 diabetes genetic risk score

The T1D GRS was calculated on the development cohort as previously described.[18] In brief, T1D GRS consists of 30 common type 1 diabetes genetic variants (single nucleotide polymorphisms (SNPs)) from HLA (Human Leukocyte Antigen) and non-HLA loci; each variant is weighted by its effect size on type 1 diabetes risk from previously published literature, with weights for DR3/DR4-DQ8 assigned based on imputed haplotypes (online supplementary table 2). All SNPs had an INFO (Information content metric)>0.8. The combined score represents an individual's genetic susceptibility to type 1 diabetes. T1D GRS calculation was not performed if genotyping results were missing for either of the two alleles with the greatest weighting (DR3/DR4-DQ8 or HLA_DRB1_15) or if more than two of any other SNPs were missing. For ease of clinical interpretation, the score is presented in this article as the score and centile position of the distribution in the Wellcome Trust Case Control Consortium type 1 diabetes population.[36]

## Statistical analysis
### Model development

We used logistic regression analysis to develop the models. Models were developed on a complete case basis.

Age at diagnosis, BMI and T1D GRS were modelled as continuous variables and transformations used to ensure linearity on the logit scale[37] (online supplementary figure 1AB). GADA and IA-2A were both dichotomised into negative or positive based on the cut-off for positivity in line with how the results are reported clinically.[2] Sample sizes were checked using both minimal events per variable (EPV) criteria (≥10)[38] and square root of the mean squared prediction error[39] and were considered sufficient for reliable diagnostic modelling.

Models were built and validated in four stages, this staged development sequence was selected in order of clinical availability of the predictors and, as some participants had missing diagnostic test data, to maximise the sample size at each stage: (1) model including only clinical features (age at diagnosis and BMI); (2) addition of GADA to the linear predictor from model 1; (3) addition of both GADA and IA-2A to the linear predictor from model 1 and (4) addition of T1D GRS to model three linear predictor.

### Evaluation of model performance: internal validation

Three internal validation techniques were used to assess the discrimination and calibration performance of the models: (1) directly using the data used to develop the model (apparent validation, area under the receiver operating characteristic curve (ROC AUC)); (2) Jack-knife

cross-validation and (3) bootstrapping (with replacement method)[37]

### Evaluation of model performance: external validation

Performances of model 1 (clinical features) and model 2 (clinical features+GADA) were evaluated in the YDX study cohort. We were unable to externally evaluate models 3 and 4 as IA-2A autoantibodies and T1D GRS were not available in the YDX study.

### Model comparisons

Four nested replica models were built on the subset of participants with complete data on all predictor variables (n=943). The predictive information of each additional predictor on the model performance was assessed using the Unitless Index of Adequacy,[37] log likelihood ratio test,[37] Net Reclassification Improvement and Integrated Discrimination Improvement.[40]

### Sensitivity analysis

Model development of all four models was repeated on 943 participants with complete data. To assess performance of biomarker models in those difficult to classify on clinical features alone, AUC ROC was repeated for each model in participants with intermediate age of diagnosis (range 25–35 years (inclusive)) and BMI (range $25–35 \, kg/m^2$ (inclusive)).

All statistical analyses were performed using STATA V.15, STATA Corp, Texas, USA (unless otherwise stated).

### Patient involvement

Patients with diabetes were involved in prioritising the research question and development of the original funding application. This study did not involve the collection of primary data, but this research was reviewed and access to data approved by the Peninsula Research Bank Lay steering committee, who also contributed to the design and development of the source cohort studies.

### RESULTS

In all, 1352 (type 1 diabetes n=179) participants met analysis inclusion criteria for the clinical features model with 943 participants having all predictor variables measured. A flow diagram describing the flow of participants through the study is shown in online supplementary figure 2. Only 37 (2.7% of the cohort) had an undefinable outcome due to intermediate C-peptide levels (200–600 pmol/L when insulin-treated diabetes within 3 years of diagnosis). The remaining exclusions were due to either missing data or short duration of diabetes. The characteristics and type 1 diabetes outcome prevalence of the included participants were similar in all four development samples (online supplementary table 3). There were no clinically relevant differences in the characteristics of the participants who were excluded from the fourth model development stage (n=409) (online supplementary table 4). Islet

autoantibodies and C-peptide were measured at median 13 years and 16 years post-diagnosis, respectively.

### Clinical features or biomarkers in isolation overlap substantially between diabetes types

Participants with type 1 diabetes and rapid insulin requirement were diagnosed younger compared to the participants with type 2 diabetes (median 27 vs 44 years, p<0.001) and had a lower BMI (median 26 vs 34 kg/m², p < 0.001). Positive autoantibodies (GADA, IA-2A or both) were more common in the participants with type 1 diabetes (71% of participants with type 1 diabetes vs 5% of participants with type 2 diabetes, p < 0.001). Patients with type 1 diabetes had a higher T1D GRS (median 0.27 vs 0.23 (equivalent to 40th and 4th centiles of the Wellcome Trust Case Control Consortium population with type 1 diabetes,[36] p < 0.001). These features overlapped substantially between participants meeting criteria for type 1 and type 2 diabetes (figure 1A–D) with AUC ROC for these features in isolation: 0.82 (age at diagnosis), 0.83 (BMI), 0.83 (islet autoantibodies) and 0.85 (T1D GRS).

### Combining clinical features using a diagnostic model improves model discrimination

In model 1, age at diagnosis and BMI were both significant independent predictors of type 1 diabetes, with the odds of having type 1 diabetes increasing with younger age at diagnosis and lower BMI. Combined, these features provided excellent discrimination (ROC AUC=0.904, perfect test=1) (figure 2A), with low probabilities capturing the majority of participants with type 2 diabetes and type 1 diabetes being very unlikely (figure 2B; sensitivity, specificity, and positive and negative predictive values at various probability cut-offs are reported in table 1). In successive models adding in GADA (model 2 (figure 2C and D)), then IA-2A (model 3 (figure 2E and F)) and then T1D GRS (model 4 (figure 2G and H)), the addition of each predictor to the previous model resulted in significant improvements in discrimination (online supplementary table 5) and model fit (online supplementary tables 67). In sensitivity analysis, results were similar when restricting all models to only the 943 participants with complete data on all predictor variables (online supplementary table 8).

In further sensitivity analysis restricting analysis to those most difficult to classify on clinical features alone due to both intermediate BMI (range $25–35 \, kg/m^2$ (inclusive)) and age of diagnosis (range 25–35 years (inclusive)), model performance remained high for models incorporating biomarker measurement (clinical features+islet autoantibodies AUC ROC 0.89, clinical features+islet autoantibodies+T1D GRS AUC ROC 0.95) (online supplementary table 9). This compares to AUC ROC of 0.72 for GADA and IA-2A measurement alone, and 0.89 for T1D GRS measurement alone in this subpopulation (n=71).

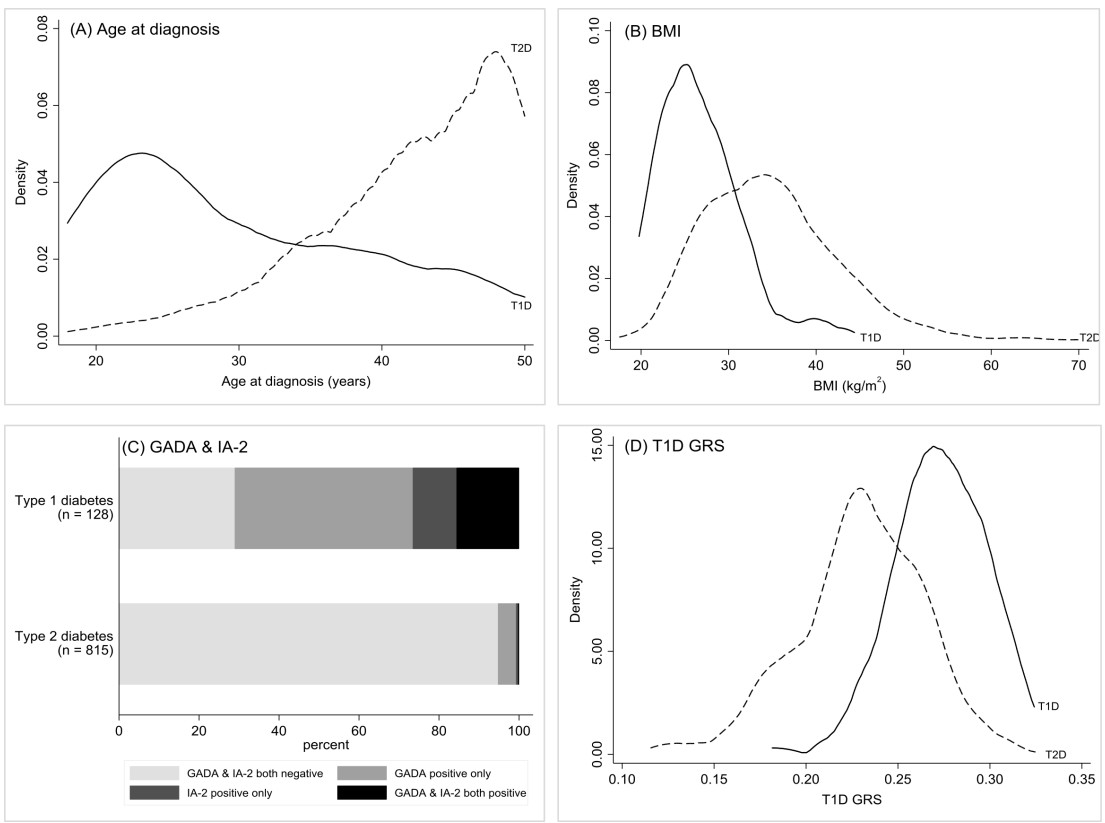

**Figure 1** Density plots for (A) age at diagnosis, (B) BMI and (D) T1D GRS. Stacked bar chart (C) showing percentages of participants (total n=943 (stage 4 model development sample)) by actual type 1 diabetes outcome and GADA/IA-2A status. Dashed line shows the distribution for T2D (n=815), solid line shows the distribution for T1D (n=128) of participants included in the stage 4 model development. BMI, body mass index; T1D GRS, type 1 diabetes Genetic Risk Score; T2D, type 2 diabetes.

### Internal validation suggests robust model performance

Results of the internal validation bootstrap (online supplementary table 5) indicate good model discrimination, with very similar model performance in bootstrapped samples (near identical ROC AUC for all models (max decrease=0.0018)), high calibration indicating the predicted probabilities closely fit the observed probabilities (calibration slope range 0.98–1.00 (0.9–1.1 is indicative of good calibration)), and very low levels of optimism suggesting little error due to overfitting.

### Model performance remains high in an external validation cohort with different characteristics

582 participants in the YDX study met criteria for external validation (Supplementary Figure 3). Compared to the participants in the Exeter model development cohort, the participants in the YDX study were younger at diagnosis (consistent with the narrower age range in YDX (18–45 years) (median 37 years vs 43 years, p < 0.001)), had a lower BMI (median 31 kg/m$^2$ vs 33 kg/m$^2$, p < 0.001), had a higher percentage of GADA (20% vs 12%, p < 0.001) and a higher prevalence of type 1 diabetes by study definition (22% vs 14%, p < 0.001) (see Supplementary Table 10 for participant characteristics).

There was a small decrease in performance of the model 1 (clinical features) and model 2 (clinical features and GADA) when they were applied to the

external validation samples but both still showed high levels of discrimination despite differences in the two cohorts (ROC AUC=0.865 and 0.930 for models 1 (figure 3A, B and C) and 2 (figure 3D, E and F), respectively (online supplementary table 11). Both models slightly over estimated type 1 diabetes prevalence but there was no evidence of miscalibration (figure 3B and E, online supplementary table 11). Sensitivity and specificity in the validation cohort are shown in online supplementary table 12.

### Participants with high model probability type 1 diabetes but type 2 diabetes outcome have the characteristics of type 1 diabetes but took > 3 years to commence insulin therapy

Online supplementary table 13 shows the characteristics of 12 participants in the external validation cohort with >80% model type 1 diabetes probability, but an actual model outcome of type 2 diabetes. These participants had the clinical characteristics associated with type 1 diabetes with GADA positivity and low C-peptide in the majority of cases (median C-peptide 120 pmol/L). However, the time to insulin was >3 years in GADA-positive cases, suggesting slow onset autoimmune diabetes. In contrast, the six participants who had a low model type 1 diabetes probability (<16%) but an actual model outcome of type 1 diabetes (online supplementary table 14) had features associated with type 2 diabetes.

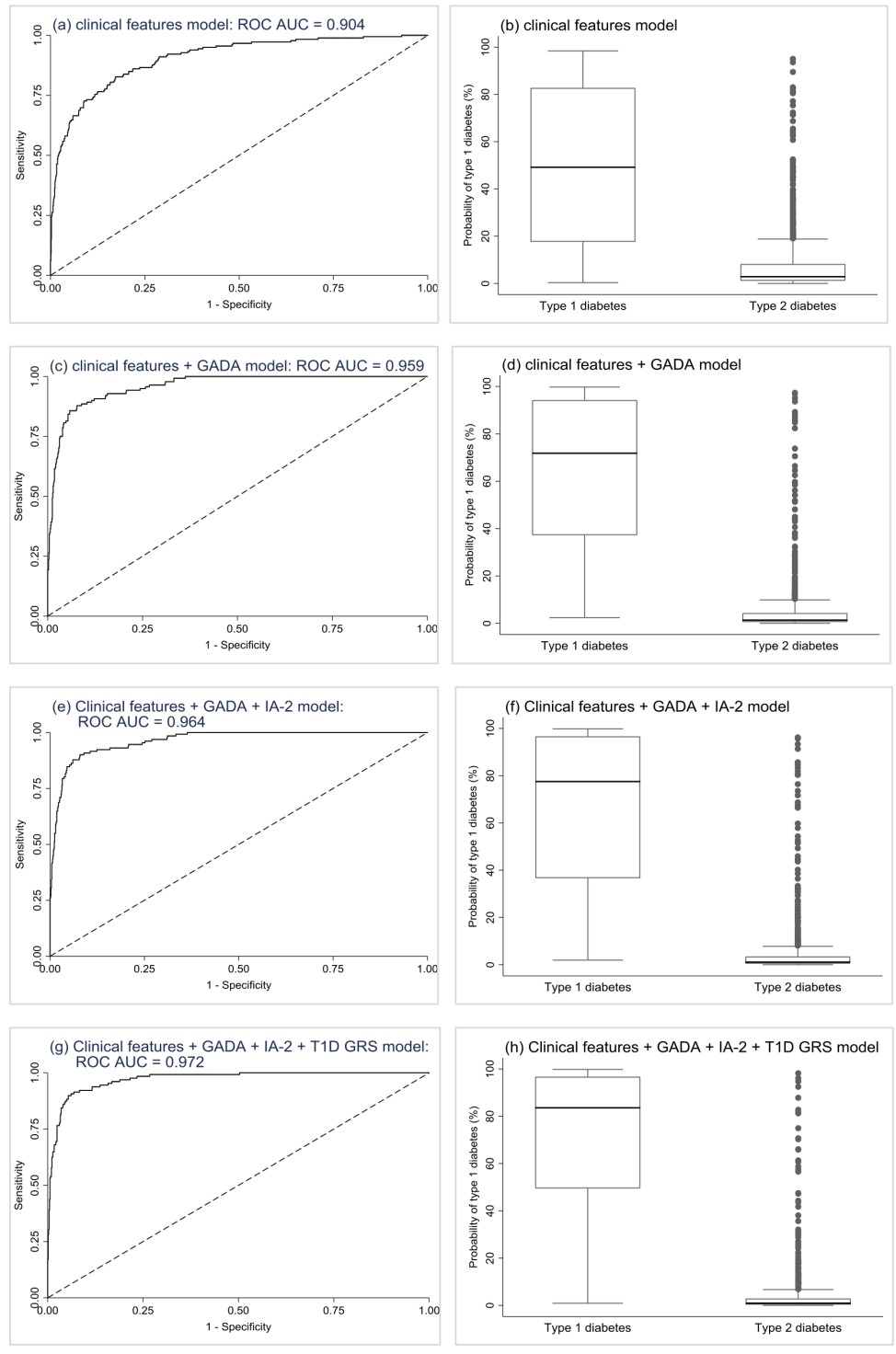

**Figure 2** Development sample validation results. Plots are the results from the validation of the models. First row (A and B): clinical features logistic regression model (n=1315). Second row (C and D): clinical features+GADA logistic regression model (n=1036). Third row (E and F): clinical features+GADA + IA-2A logistic regression model (n=1025). Fourth row (G and H): clinical features+GADA + IA-2A+T1D GRS logistic regression model (n=943). Plots (A), (C), (E), & (G) are ROC curves showing discrimination ability of the models. Plots (B), (D), (F) and (H) are boxplots of fitted model probabilities grouped by actual diabetes outcome. ROC, receiver operating characteristic; T1D GRS, Type 1 Diabetes Genetic Risk Score.

## Online calculator

The four models have been incorporated into an online calculator (beta version available at https://www.diabetes-genes.org/t1dt2d-prediction-model/). An additional four models with different combinations of the five predictor variables were also developed for the online calculator, to allow every combination of clinical features plus the other biomarkers as optional. As expected, ROC AUC and prediction error results for these four additional models were intermediate between the basic clinical features model and

**Table 1** Model performance at different cut-offs for all four logistic regression models (development cohort). Positive and negative predictive values relate to type 1 diabetes NPV, negative predictive value; PPV, positive predictive value

**Clinical features (n=1352)**

| | Probability (%) cut-off for classifying type 1 diabetes | | | | | |
| --- | --- | --- | --- | --- | --- | --- |
| | 10 | 30 | 50 | 70 | 90 | 12 (Youden's Index) |
| Sensitivity/specificity (%) | 85/79 | 64/95 | 49/98 | 35/99 | 15/100 | 83/83 |
| Accuracy (%) | 80 | 90 | 91 | 90 | 89 | 83 |
| PPV (%) | 38 | 64 | 79 | 83 | 90 | 42 |
| NPV (%) | 97 | 95 | 93 | 91 | 89 | 97 |

**Clinical features+GADA (n=1036)**

| | Probability (%) cut-off for classifying type 1 diabetes | | | | | |
| --- | --- | --- | --- | --- | --- | --- |
| | 10 | 30 | 50 | 70 | 90 | 16 (Youden's Index) |
| Sensitivity/specificity (%) | 90/88 | 80/96 | 66/97 | 52/99 | 31/100 | 86/92 |
| Accuracy (%) | 89 | 94 | 93 | 92 | 90 | 92 |
| PPV (%) | 55 | 75 | 80 | 85 | 92 | 64 |
| NPV (%) | 98 | 97 | 95 | 93 | 90 | 98 |

**Clinical features+GADA + IA-2A (n=1025)**

| | Probability (%) cut-off for classifying type 1 diabetes | | | | | |
| --- | --- | --- | --- | --- | --- | --- |
| | 10 | 30 | 50 | 70 | 90 | 12 (Youden's Index) |
| Sensitivity/specificity (%) | 91/91 | 80/96 | 69/98 | 57/99 | 37/100 | 90/92 |
| Accuracy (%) | 91 | 94 | 94 | 93 | 92 | 92 |
| PPV (%) | 59 | 75 | 81 | 85 | 92 | 62 |
| NPV (%) | 99 | 97 | 96 | 94 | 92 | 98 |

**Clinical features+GADA + IA-2A+T1D GRS (n=943)**

| | Probability (%) cut-off for classifying type 1 diabetes | | | | | |
| --- | --- | --- | --- | --- | --- | --- |
| | 10 | 30 | 50 | 70 | 90 | 14 (Youden's Index) |
| Sensitivity/specificity (%) | 92/90 | 84/96 | 74/98 | 63/99 | 41/100 | 91/93 |
| Accuracy (%) | 90 | 95 | 94 | 94 | 92 | 93 |
| PPV (%) | 59 | 78 | 83 | 88 | 93 | 67 |
| NPV (%) | 99 | 98 | 96 | 94 | 92 | 99 |

Accuracy = (true positives + true negatives)/total number of participants. PPV = [(sensitivity × prevalence)/[(sensitivity × prevalence) + ([1 − specificity] × [1 − prevalence])]. NPV = [specificity × (1 − prevalence)]/[(specificity × [1 − prevalence]) + ([1 − sensitivity] × prevalence)]. Youden's Index − best trade-off between sensitivity and specificity (sensitivity+specificity − 1).
NPV, negative predictive value; PPV, positive predictive.

the full model with all features (see online supplementary table 15).

Online supplementary tables 16–23 inclusive show the β-coefficients and ORs for all models. The regression equations for the online calculator are shown in online supplementary table 24.

## DISCUSSION

We have developed, evaluated and validated clinical diagnostic models combining age at diagnosis, BMI, GADA, IA-2 and T1D GRS to provide estimates of a patient's risk of having type 1 diabetes requiring rapid insulin therapy from diagnosis. These models show high performance and could potentially assist classification of diabetes in clinical practice and provide a tool for evidence-based classification in research cohorts.

Model performance was optimised in the model combining all five predictors (ROC AUC 0.97). However, all models performed well with ROC AUC >0.9 and low cross-validated prediction errors in development. The results of the external validation provide additional confidence in model performance. This was undertaken in a distinct dataset with different type 1 diabetes prevalence and biochemical assays.

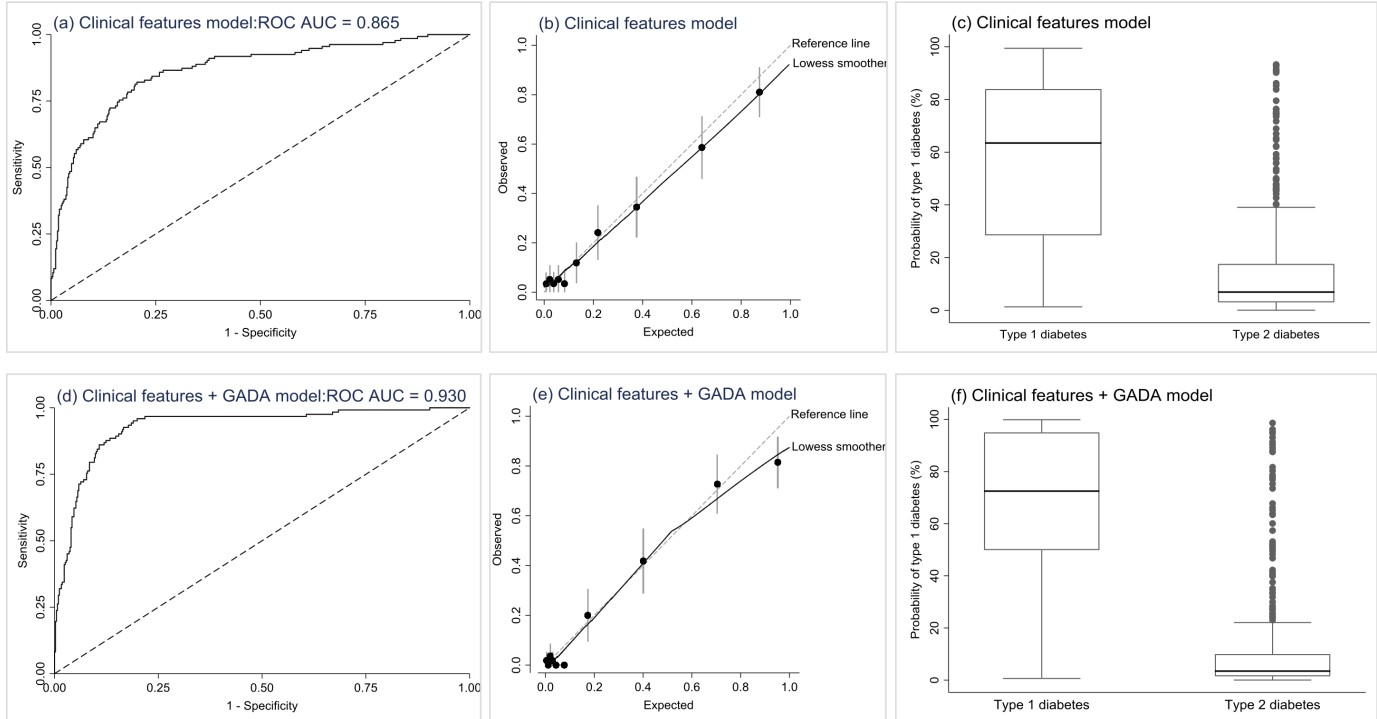

**Figure 3** External validation results. Plots on the first row (a, b, c) are the results from the external validation of the clinical features logistic regression model applied to participants in the YDX study (n=582). The second row of plots (d, e and f) are the results from the external validation of the clinical features+GADA logistic regression model applied to participants in the YDX study (n=549). Plots (a) and (d) are ROC curves showing discrimination ability of the models, dashed line represents the reference line. Plots (b) and (e) are calibration plots. Plots (c) and (f) are boxplots of fitted model probabilities grouped by actual diabetes outcome. ROC, receiver operating characteristic; YDX, Young Diabetes in Oxford.

This is the first study developing clinical diagnostic models for classification of type 1 and 2 diabetes. Key strengths of this study include our systematic approach to model development including robust internal and external validation.[41] Our staged approach to model development means that we have maximised the information gained from each predictor. Our model is parsimonious, we have used only five predictors previously shown to be associated with type 1 diabetes. This, in combination with large datasets, mean we have a high number of EPV and very low risk of overfitting, a common problem with diagnostic models of this nature. Our use of predominantly population-based cohorts recruited largely from a primary care setting (for model development) means our results are likely to reflect true associations in patients seen in clinical practice. The overall prevalence of study defined type 1 diabetes of 13% in our development dataset is close to the 11% reported type 1 diabetes prevalence at diagnosis in a UK population aged 20–50 years.[42]

A limitation of our study is the cross-sectional nature of our cohorts meaning that age at diagnosis and time to insulin were self-reported at a single visit. Insulin commencement was also based on clinical decision-making rather than a trial protocol. BMI and antibodies were measured at median 13 years after diagnosis. BMI, and GAD and IA-2A antibodies change modestly over time in adult onset diabetes, with

previous research suggesting an approximately 18% lower combined GADA and IA-2A prevalence after 13.5 years diabetes duration in this age group,[43] and BMI having higher discrimination for diabetes classification when measured at diagnosis.[44] The potential impact on the results of BMI and islet autoantibodies having been measured some years post-diagnosis is that the predictions may be under-estimated. The lack of information at diagnosis also meant we were unable to assess whether other features available at diagnosis may assist classification, such as presentation glycaemia, ketosis or weight loss. A prospective study to validate these models, and assess whether other features may assist classification is therefore ongoing (https://clinicaltrials.gov/ct2/show/NCT03737799).

A further limitation is that this model has been developed and tested in a white European population with young onset diabetes, extension of this work to non-white populations and older age groups is therefore a priority for future research.

These models have the potential to help robustly classify diabetes in research cohorts, and may have particular utility where genetic but not antibody data is available, a common situation in many biobanks. They may also assist clinical decision-making, with the important caveats that this evidence can only be applied to patients aged 18–50 years, of white ethnicity, and that these models are intended to act as a decision aid in conjunction with

other information which a clinician may use to inform treatment decisions (eg, severity of hyperglycaemia): they do not replace expert clinical opinion. A web-based calculator and smartphone app could be used to display the estimate of the patient's probability of having type 1 diabetes based on the predictor variable values entered. The models can be used with age of diagnosis and BMI as a minimum; users will then have a choice to add results of GADA, IA-2A and T1D GRS in any combination. This could therefore be used by clinicians as a triage-based approach to diabetes subtype diagnosis. For example, probabilities calculated on clinical features could be used as the basis for antibody testing, or the additional value likely to be gained from antibody or genetic testing could be assessed by inputting dummy results into the model. We propose providing the continuous probability outcome of the models rather than giving a threshold. This is because the decision made on whether to commence insulin for a given probability of type 1 diabetes will vary enormously due to other factors. For example, temporary insulin treatment may be appropriate regardless of likely classification where hyperglycaemia is severe, and in some circumstances it may be appropriate to trial oral therapy even where type 1 diabetes has a high probability, for example where a person's occupation would be affected by insulin treatment and they can be carefully monitored for glycaemic deterioration.

In conclusion, clinical diagnostic models integrating clinical features with biomarkers have high accuracy for identifying type 1 diabetes with rapid insulin requirement in white participants aged 18–50 years at diabetes diagnosis, and may assist clinicians in identifying patients with type 1 diabetes in clinical practice.

**Author affiliations**
[1]The Institute of Biomedical & Clinical Science, University of Exeter Medical School, Exeter, UK
[2]Department of Clinical Biochemistry, Royal Devon and Exeter NHS Foundation Trust, Exeter, UK
[3]Kidney Unit, Royal Devon and Exeter NHS Foundation Trust, Exeter, UK
[4]Molecular and Clinical Medicine, University of Dundee, Dundee, UK
[5]Macleod Diabetes and Endocrine Centre, Royal Devon and Exeter NHS Foundation Trust, Exeter, UK
[6]Oxford Centre for Diabetes Endocrinology and Metabolism, University of Oxford, Oxford, UK
[7]Oxford NIHR Biomedical Research Centre, Oxford University Hospitals Foundation Trust, Oxford, UK

**Acknowledgements** The authors thank participants who took part in these studies and the research teams who undertook cohort recruitment. We thank Catherine Angwin of the NIHR (National Institute of Health Research) Exeter Clinical Research Facility for assistance with data preparation, and Rachel Nice of the Blood Sciences Department, Royal Devon and Exeter Hospital for assistance with sample analysis. We are grateful to Maarten van Smeden for allowing us to access to his Beyond EPV R-Shiny app (BETA version).

**Contributors** ALL, BMS and AGJ conceived the idea and designed the study. ALL, TJM, AVH, ERP, MNW, ATH, KRO and AGJ researched the data. ALL analysed the data with assistance from BMS and AGJ. TJM, JMD, RAO, ATH and KRO discussed and contributed to study design and provided support for the analysis and interpretation of results. ALL drafted the manuscript with assistance from BMS and AGJ. All authors critically revised the manuscript and approved the final version. The corresponding author attests that all listed authors meet authorship criteria and that no others meeting the criteria have been omitted.

**Funding** NIHR Oxford Biomedical Research Centre. The Diabetes Alliance for Research in England (DARE) study was funded by the Wellcome Trust and supported by the Exeter NIHR Clinical Research Facility. The MASTERMIND study was funded by the UK Medical Research Council (MR/N00633X/) and supported by the NIHR Exeter Clinical Research Facility. The PRIBA study was funded by the National Institute for Health Research (NIHR) (UK) (DRF-2010-03-72) and supported by the NIHR Exeter Clinical Research Facility. TJM is a National Institute for Health Research Senior Clinical Senior Lecturer. RAO is supported by a Diabetes UK Harry Keen Fellowship (16/0005529). ERP is a Wellcome Trust New Investigator (102820/Z/13/Z). ATH and BMS are supported by the NIHR Exeter Clinical Research Facility. ATH is a Wellcome Trust Senior Investigator and NIHR Senior Investigator. AGJ is supported by an NIHR Clinician Scientist award (CS-2015-15-018). This publication presents independent research funded by the National Institute for Health Research (NIHR). The views expressed are those of the author and not necessarily those of the NHS, the NIHR or the Department of Health and Social Care. The study sponsor was not involved in the design of the study; the collection, analysis and interpretation of data; writing the report; or the decision to submit the report for publication.

**Competing interests** None declared.

**Patient consent for publication** Not required.

**Ethics approval** All cohort studies used for this research received ethical approval from the UK National Research Ethics Service. All participants gave written informed consent.

**Provenance and peer review** Not commissioned; externally peer reviewed.

**Data availability statement** Data from the Exeter cohorts included in this research is held by the Peninsula Research Bank, managed by the NIHR Exeter Clinical Research Facility. Guidance for applying to use the Peninsula Research Bank resource are given on the following website: https://exetercrfnihr.org/about/exeter-10000-prb

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
