## [Reviewer comments · BMJ Open]

ARTICLE DETAILS

TITLE (PROVISIONAL)	Development and validation of multivariable clinical diagnostic models to identify type 1 diabetes requiring rapid insulin therapy in adults aged 18 to 50
AUTHORS	Lynam, Anita; McDonald, Timothy; Hill, Anita; Dennis, John; Oram, Richard; Pearson, Ewan; Weedon, Michael; Hattersley, Andrew; Owen, Katharine; Shields, Beverley; Jones, Angus

VERSION 1 - REVIEW

REVIEWER	Taulant Muka Institute of Social and Preventive Medicine, University of Bern, Switzerland
REVIEW RETURNED	05-Jun-2019

GENERAL COMMENTS	Lynam et al and colleagues, using data from more than 1900 diabetes patients investigate whether 5 selected clinical variables could help to distinguish T1D and T2D in adult age. Authors, based on the five selected predictors, develop and replicate a prediction model with high accuracy for identifying type 1 diabetes Comments I have a concern with the terms authors use. Considering this is a cross-sectional study, would be more appropriate to use the term diagnostic model than prediction model. Participants selected for the analysis have been for many years diagnosed with diabetes. Would this have an impact in the results considering the cross-sectional design? Authors need to elaborate on this. Why did author select only five variables and no other biomarkers such as inflammatory biomarkers? Also, did authors have information on family history for diabetes? Why using only the genetic risk score for T1D and not including the genetic risk score for T2D as well?
---

REVIEWER	Antonio Martinez-Millana Universitat Politècnica de València, Spain
REVIEW RETURNED	05-Jun-2019

GENERAL COMMENTS	This paper describes a comprehensive study aiming at developing and validating a multivariable model to classify type 1 and type 2 diabetes from a pool of patients diagnosed with diabetes. Even though the paper is large, it is well written, concise and consistent (statistics are very well reported). Main concerns about the paper are on the methodological approach and the data set used to develop and validate the models. 1) First is about the problem authors want to address and the background they describe in the introduction. Authors state that there are no models to help distinguish T1 and T2 at diagnosis, but this is untrue as for the references presented before. According to Ref 10 Age at the diagnose (years with diabetes) and BMI are no longer basis for classifying the disease type. Moreover, ref 18 suggests that GRS can accurately identify young adults with diabetes who will require insulin treatment but may need BMI/Age at diagnosis for values under a threshold. Authors chose a step-wise method based on clinical variables (Years since diagnose, BMI), and addition of GADA, IA-2A and GSR. What are the basis for choosing these predictors and this model development sequence? Why did authors not include C-peptide? A comparison of existing approaches to identify T1/T2, predictors, model development characteristics and performance can help to reinforce the problem statement. 2) Time is a crucial dimension in the ascertainment of T1D or T2D. Importantly authors state as a limitation the cross-sectional nature of the majority of the data used in this study, but based on these limitations, the criteria for the outcome definition seems to be weak and overlapping and the pre-processing of collected c-peptides (68%) may have influenced this. 3) Figure 1 reveals that model 1 will yield a good performance, as the two type of outcomes are clearly differentiated. Have authors considered working only in the misclassification areas? This is, when the age at diagnosis is from 32 to 36, when the BMI is around 31. It is in these cases when clinical variables may not be good predictors, as stated in ref 10. 4) Authors should indicate when the BMI variable was collected. T1D is characterized by a severe decrease of the BMI in the moment of the diagnose, whereas T2D is the opposite. Minor: 1) Consider re-writing the first paragraph of the Results, it is hard to follow without the supplemental figure.
---

VERSION 1 – AUTHOR RESPONSE

	Comment	Response
Reviewer: 1	I have a concern with the terms authors use. Considering this is a cross-sectional study, would be more appropriate to use the	We have stated in the introduction that our model is a diagnostic prediction model as per the TRIPOD guidelines. We have amended the title and manuscript to include the term diagnostic where we refer to the model as a prediction model to make the medical context clearer.

	term diagnostic model than prediction model.	
Reviewer: 1	Participants selected for the analysis have been for many years diagnosed with diabetes. Would this have an impact in the results considering the cross-sectional design? Authors need to elaborate on this.	The potential impact on the results of BMI and islet-autoantibodies having been measured some years post diagnosis is that the predictions may be under-estimated. The collection of predictors post-diagnosis is discussed as a limitation of our study in the discussion section, we have added additional text to further emphasize this (page 20).
Reviewer: 1	Why did author select only five variables and no other biomarkers such as inflammatory biomarkers? Also, did authors have information on family history for diabetes?	We pre-selected variables based on previous evidence of association with diabetes subtype classification. We did not have data on the suggested markers (C-reactive protein, Interleukin-6, Interleukin-8, Transforming growth factor β1, Monocyte chemotactic protein 1, Leptin, Adiponectin) and to our knowledge these markers do not have evidence of clinical utility in diabetes classification at present. This would be an interesting area for future research. While the datasets used contained some information on first degree relative diabetes (though not the subtype of the family member) both type 1 and type 2 diabetes have a genetic component and there is very limited previous evidence that family history is a useful differentiating factor. We did explore this variable during the model build, it was predictive in isolation but missing in circa 25% of cases and added minimally to other features therefore it was not included in the model.
Reviewer: 1	Why using only the genetic risk score for T1D and not including the genetic risk score for T2D as well?	We pre-selected predictors based on previous evidence. The T2D genetic risk score was not pre-selected as we have previously shown that a 69 SNP type 2 diabetes genetic risk score has no utility in discriminating patients with type 1 from type 2 diabetes (Oram 2016).
Reviewer: 2 Comment: 1	First is about the problem authors want to address and the background they describe in the introduction. Authors state that there are no models to help distinguish T1 and T2 at diagnosis, but this is untrue as for the references presented before. According to Ref 10 Age at the diagnose (years with diabetes) and	The references presented before are either clinical guidelines or genetic risk scores relating to individual use of our predictors in diabetes classification. Our statement "there are no models to help distinguish T1 and T2 diabetes at diagnosis" is referring to statistical prediction models which do not currently exist. We have amended the text in the introduction to clarify our use of the word 'models' in this specific context to avoid confusion (page 8). We agree with Palmer's statement (Ref 10) that conventional criteria is no longer sufficient to classify patients. There is however very clear

	BMI are no longer basis for classifying the disease type. Moreover, ref 18 suggests that GRS can accurately identify young adults with diabetes who will require insulin treatment but may need BMI/Age at diagnosis for values under a threshold.	evidence that age and BMI do have utility – indeed alongside rapid insulin requirement these are the only clinical features that have robust evidence for discrimination on systematic review (Shields, 2015). Ref 18 (Oram, 2016) suggests that GRS can accurately identify young adults with diabetes who will require insulin treatment but it provides imperfect discrimination in isolation – there is marked overlap between the T1DGRS levels seen in type 1 and 2 diabetes. As our article shows age and BMI when combined markedly outperform the performance of T1DGRS or antibodies used in isolation. We use Palmer’s statement and this limitation of the T1D GRS to support the need for multivariable models in diabetes classification. In our model T1D GRS is used in combination with age at diagnosis and BMI as a minimum – as our data show this has markedly better performance than T1DGRS used alone.
Reviewer: 2 Comment: 1	Authors chose a step-wise method based on clinical variables (Years since diagnose, BMI), and addition of GADA, IA-2A and GSR. What are the basis for choosing these predictors and this model development sequence? Why did authors not include C-peptide? A comparison of existing approaches to identify T1/T2, predictors, model development characteristics and performance can help to reinforce the problem statement.	We pre-selected the five variables based on previous evidence of association with diabetes subtype classification and data availability (model predictors subsection of methods). This includes the systematic review which we have previously undertaken to assess evidence for clinical features (Shields 2015). The model sequence was chosen in order of clinical availability (BMI and age of diagnosis are available at no cost, islet antibodies are widely available but moderately expensive (some laboratories only offer GAD), at present T1DGTRS has limited availability) and to maximize the number of patients in the model build datasets. In the supplementary material and online calculator, all combinations of predictors are available. We have updated the text relating to the model build sequence in the model development subsection of methods, to make this clearer (page 13). We did not include c-peptide as a predictor in the model as it has limited clinical utility at diagnosis - insulin secretion may be preserved at type 1 diabetes diagnosis, particularly in the obese, therefore levels may overlap between type 1 and type 2 diabetes (Jones (2013), PMID 23413806). Comparing with existing approaches is a challenge as there are almost no guidelines for diabetes classification. Where attempts have been made to give advice they are vague (usually phrased along the lines of ‘consider type 1 diabetes if’ with a long list of possibly associated

		features) and therefore can-not be coded in a dataset. We have shown in the article that all of these models out perform use of islet antibodies or T1DGRS alone to classify diabetes. We have now also added performance verses islet autoantibodies or T1DGRS alone in those who are difficult to classify on clinical features alone (see below – results page 17 last paragraph).
Reviewer: 2 Comment: 2	Time is a crucial dimension in the ascertainment of T1D or T2D. Importantly authors state as a limitation the cross-sectional nature of the majority of the data used in this study, but based on these limitations, the criteria for the outcome definition seems to be weak and overlapping and the pre-processing of collected c-peptides (68%) may have influenced this.	We acknowledge that the cross-sectional nature of the predictor data used in this study is a limitation and have discussed the impact of this limitation on the results. We disagree strongly with the comment that our outcome definition is weak or overlapping. Firstly there was no overlap in outcome – this was robustly defined based on loss (or preservation) of endogenous insulin secretion (C-peptide) and early insulin requirement, with no overlap between definitions and only a tiny proportion of participants (2%) not classifiable (due to intermediate C-peptide). There is overwhelming evidence that differences in glycaemic treatment requirement and response between type 1 and type 2 diabetes are driven by the development of severe insulin deficiency (as measured by C-peptide) in the latter. Therefore direct measurement of endogenous insulins secretion in longstanding diabetes provides a very robust outcome that (in contrast to using islet antibodies or genetics for example) directly determines treatment requirements. We do not understand the comment regarding pre-processing of collected C-peptides as 68% does not appear in the manuscript. There was no pre-processing of C-peptide. Repeat values were available in 62% of patients which is a strength of the study, the median values was used in these cases.
Reviewer: 2 Comment: 3	Figure 1 reveals that model 1 will yield a good performance, as the two type of outcomes are clearly differentiated. Have authors considered working only in the misclassification areas? This is, when the age at	Age at diagnosis and BMI alone will give intermediate predictions in those where both age and BMI are intermediate. This is the patient group where additional testing is likely to have greatest utility. Where clinical features result in intermediate probabilities the clinical features model would indicate intermediate probability, and users have the choice to then add results of autoantibodies and genetic testing to improve

	diagnosis is from 32 to 36, when the BMI is around 31. It is in these cases when clinical variables may not be good predictors, as stated in ref 10.	predictions. To illustrate the utility of the models that incorporate antibodies and/or type 1 diabetes genetic risk score in those where clinical features are intermediate we have now assessed model performance in those with age of diagnosis and BMI are both intermediate (Page 17). For this analysis we used somewhat broader criteria than suggested (participants with diagnosis age 25-35 and BMI 25-35) due to numbers with these characteristics in the dataset.
Reviewer: 2 Comment: 4	Authors should indicate when the BMI variable was collected. T1D is characterized by a severe decrease of the BMI in the moment of the diagnose, whereas T2D is the opposite.	BMI was measured at median 13 years post diagnosis, this is included in the discussion (page 21 paragraph 2) where we discuss limitations of the cross sectional nature of this dataset, it also corresponds to the duration of diabetes at recruitment (patient characteristics - supplementary table 3). We have not included weight loss as a predictor as it does not have an evidence base (see Shields 2015 PMID: 26525723) and we did not have this data available. The lack of information at diagnosis (including weight loss) is discussed in the discussion section with reference to an ongoing prospective study which will address this limitation.
Reviewer: 2 Minor Comment: 1	Consider re-writing the first paragraph of the Results, it is hard to follow without the supplemental figure.	We have removed the sentence on initial treatment from this paragraph so that this now relates entirely to model outcome (page 16).

VERSION 2 – REVIEW

REVIEWER	Taulant Muka Institute of Social and Preventive Medicine, University of Bern, Switzerland
REVIEW RETURNED	25-Jul-2019

GENERAL COMMENTS	Authors have successfully addressed my comments, no further comments.
---